# Nutritional Genomic Approach for Improving Grain Protein Content in Wheat

**DOI:** 10.3390/foods12071399

**Published:** 2023-03-25

**Authors:** Tania Kartseva, Ahmad M. Alqudah, Vladimir Aleksandrov, Dalia Z. Alomari, Dilyana Doneva, Mian Abdur Rehman Arif, Andreas Börner, Svetlana Misheva

**Affiliations:** 1Institute of Plant Physiology and Genetics, Bulgarian Academy of Sciences, Acad. G. Bonchev Str., Block 21, 1113 Sofia, Bulgaria; tania_karceva@abv.bg (T.K.); aleksandrov@gbg.bg (V.A.); donevadiliana@gmail.com (D.D.); 2Biological Science Program, Department of Biological and Environmental Sciences, College of Art and Science, Qatar University, Doha P.O. Box 2713, Qatar; aalqudah@qu.edu.qa; 3Department of Clinical Nutrition and Dietetics, Faculty of Applied Medical Sciences, The Hashemite University, P.O. Box 330127, Zarqa 13133, Jordan; daliaz.alomari@hu.edu.jo; 4Wheat Breeding Group, Plant Breeding and Genetics Division, Nuclear Institute for Agriculture and Biology (NIAB), Faisalabad 38000, Pakistan; m.a.rehman.arif@gmail.com; 5Leibniz Institute of Plant Genetics and Crop Plants Research (IPK Gatersleben), Corrensstraße 3, OT Gatersleben, 06466 Seeland, Germany; boerner@ipk-gatersleben.de

**Keywords:** association mapping, bread wheat, candidate genes, micronutrients, grain proteins

## Abstract

Grain protein content (GPC) is a key aspect of grain quality, a major determinant of the flour functional properties and grain nutritional value of bread wheat. Exploiting diverse germplasms to identify genes for improving crop performance and grain nutritional quality is needed to enhance food security. Here, we evaluated GPC in a panel of 255 *Triticum aestivum* L. accessions from 27 countries. GPC determined in seeds from three consecutive crop seasons varied from 8.6 to 16.4% (11.3% on average). Significant natural phenotypic variation in GPC among genotypes and seasons was detected. The population was evaluated for the presence of the trait-linked single nucleotide polymorphism (SNP) markers via a genome-wide association study (GWAS). GWAS analysis conducted with calculated best linear unbiased estimates (BLUEs) of phenotypic data and 90 K SNP array using the fixed and random model circulating probability unification (FarmCPU) model identified seven significant genomic regions harboring GPC-associated markers on chromosomes 1D, 3A, 3B, 3D, 4B and 5A, of which those on 3A and 3B shared associated SNPs with at least one crop season. The verified SNP–GPC associations provide new promising genomic signals on 3A (SNPs: *Excalibur_c13709_2568* and *wsnp_Ku_c7811_13387117*) and 3B (SNP: *BS00062734_51*) underlying protein improvement in wheat. Based on the linkage disequilibrium for significant SNPs, the most relevant candidate genes within a 4 Mbp-window included genes encoding a subtilisin-like serine protease; amino acid transporters; transcription factors; proteins with post-translational regulatory functions; metabolic proteins involved in the starch, cellulose and fatty acid biosynthesis; protective and structural proteins, and proteins associated with metal ions transport or homeostasis. The availability of molecular markers within or adjacent to the sequences of the detected candidate genes might assist a breeding strategy based on functional markers to improve genetic gains for GPC and nutritional quality in wheat.

## 1. Introduction

Bread wheat (*Triticum aestivum* L.) is the staple crop for an estimated 35–40% of the world population and provides more calories and protein in the human diet than any other crop [1]. Grain proteins are a major determinant of the end-use quality and an important nutritional trait of wheat [2]. Besides being a primary source of protein, wheat is also a major supply of other essential and beneficial compounds for human health including starch, minerals, vitamins, and fibers. Hence, wheat is a decisive crop for global food security, and increases in grain production are required to feed the expanding human population. Tremendous gains in wheat production have occurred since the 1960s due to the introduction of semi-dwarf, lodging-resistant, and nitrogen-responsive wheat lines [3]. However, the growing demands of consumers for healthier and higher-nutritional-value food have shifted the focus of breeding programs on improving grain quality in addition to high productivity. Moreover, malnutrition is recognized as a serious global health issue, due to the deficiency in proteins and micronutrients. The ongoing appeal for optimal nutrition is also a driver for the food industry. For a while, innovative fortified wheat such as bakery products with improved protein and enhanced overall nutritional contents are being introduced as a short-term solution [4,5]. However, the world is looking for a sustained science-based solution relying on genetic biofortification to address the human nutritional requirements [6,7,8]. Such nutritional genetic approaches necessitate the identification of responsible genes and their deployment for developing improved crops [9].

Mature wheat grain contains 8 to 20% proteins (10–12% on average) [10]. Grain proteins are localized in the starchy endosperm, aleurone layer, seed coat and embryo, and are classified into three groups by their function: storage proteins (gliadins, glutenins, and certain albumins and globulins), structural and metabolic proteins, and protective proteins [11]. Grain protein content (GPC) is a quantitative trait predisposed by a complex interplay between genes and environment. A large number of genomic regions or quantitative trait loci (QTL) underlying GPC had been reported on many chromosomes. Often, narrow genetic resources and the high dependance of GPC on environmental factors are the main drawbacks that limit the detection of stable GPC-QTLs. Most of what we know about the genetic architecture of GPC relies on bi-parental populations (reviewed in [12]). In bread wheat, for example, thirteen QTL on chromosomes 2A, 2B, 2D, 3D, 4A, 6B, 7A, and 7D have been reported in a mapping population of 100 recombinant inbred lines (RILs) derived from a cross between a low-protein and a high-protein parents [13]. Other studies focused on durum wheat to explore the genetic factors of GPC. For instance, on the basis of a doubled haploid Canadian durum wheat population consisting of 162 lines derived from Pelissier × Strongfield cross and 1212 polymorphic single nucleotide polymorphisms (SNPs), four QTL (*QGpc.spa-2B.1*, *QGpc.spa-2B.2*, *QGpc.spa-3A.3*, and *QGpc.spa-7A*) have been consistently detected in multiple environments [14].

In contrast, very few of the research on GPC genetics is based on genome-wide association studies (GWAS). Association mapping is a complementary approach to bi-parental linkage analysis and searches for functional polymorphisms at many loci across a set of diverse germplasm providing higher mapping resolution [15]. GWAS has been used to identify the genetic factors behind the natural variation in different diverse populations including tetraploid and hexaploid wheat [16,17,18,19]. For instance, GWAS revealed novel genomic regions associated with high GPC in 161 wild emmer wheat lines on chromosomes 2B (*QGpc.cd1-2B.1*) and 7B (*QGpc.cd1-7B.2*) using 13,116 DArT-seq markers [16]. Using a population of around 800 spring bread wheat accessions genotyped by 5095 SNPs, significant markers associated with GPC were detected on chromosomes 2A, 3B, 3D, 5B, 6B, and 7B [17]. Nine significant GPC-associated markers had been physically mapped to chromosome 6A in 93 spring wheat cultivars using GWAS with 9351 SNPs [18]. Therefore, multi-environment screening of new large populations is an imperative need to detect novel stable QTLs. Exploring new GPC genomic regions and their deployment to enhance grain protein is one of the most effective approaches for wheat quality breeding. 

Additionally, many conventional QTL studies or GWAS suggest that the accumulation of protein and other components essential for human health such as minerals are sharing common genomic regions. This may contribute to their more efficient manipulation for improving wheat grain quality, thereby mitigating global micronutrient and protein malnutrition [9,19]. For instance, the wild emmer allele *Gpc-B1* on chromosome 6B derived from *T. dicoccoides* is effective in improving grain protein, Zn and Fe concentrations [20]. QTLs conferring protein content that share loci with high Zn and/or Fe have been reported on wheat chromosomes including 2A, 2B, 5A, 6B and 7B [21,22]. Such pleiotropic regions harboring co-located QTL for grain protein and micronutrient contents were recently identified on chromosomes 3A, 3B, 5A and 7D [9,23,24].

The present study aims to characterize the natural phenotypic variation for GPC in field-grown samples in a diverse panel of 255 bread wheat accessions originating from 27 countries in 5 continents, and to identify genomic regions and candidate genes associated with GPC through GWAS. Here, we found high natural variation in GPC in wheat plants and the most promising genomic intervals underlying the variation. Furthermore, we discussed the findings in the context of the genetic relations between grain protein content and micronutrient accumulation in wheat grains. Linking protein content of wheat grain with responsible genomic regions and their candidate genes will provide an extensive insight into the potential route of improving GPC and the overall nutritional quality of wheat. 

## 2. Materials and Methods

### 2.1. Plant Material

A worldwide diverse collection of 255 winter wheat (*Triticum aestivum* L.) accessions was used for GPC and association mapping analyses. The population consisted of 192 advanced cultivars, 53 breeding lines, and 10 doubled haploid lines, originating from 27 countries on 5 continents (Appendix A). One part of the entries was selected based on preexisting information about their performance under different growing conditions during wintertime. In addition, representatives of the core collection of the Institute of Field and Vegetable Crops (IFVCNS), Novi Sad, Serbia, and parental lines of Western European breeding programs were also included. Seed samples were provided by the Seed Genebank, Leibniz Institute for Plant Genetics and Crop Plants Research (IPK, Gatersleben), Germany. The seed material was harvested from a three-growing-season field experiment (harvest 2016, 2017, and 2018) performed in Gatersleben, Germany (51°49′ N, 11°16′ E), in a random design in double rows and two replications for each accession and each growing season. Information on the monthly weather statistics for the region of the experimental field is given in Appendix A.

### 2.2. Preparation of Grain Samples and Measurement of GPC

Seeds were dried to a constant weight and ground into fine powder with IKA Tube Mill Control (IKA Werke GmbH & Co, Staufen, Germany). For each accession and harvest year, three whole-grain powder samples (1.0 g each) were measured. The total nitrogen (N) concentration was determined via the Kjeldahl method using the UDK 159 Automatic Kjeldahl Nitrogen Protein Analyzer (Velp Scientifica, Usmate Velate, Italy). GPC (%) was calculated by multiplying grain N concentration by 5.62 according to [25].

### 2.3. Statistical Analysis

The phenotypic data were analyzed (by ANOVA) to check the significant differences in GPC among accessions, environments (growing seasons) and to test the interaction effects. The best linear unbiased estimates (BLUEs) for each accession across the growing seasons were obtained to eliminate the environmental impact by assuming the genotype as a fixed effect and the growing season as a random effect using the fixed and random model circulating probability unification (FarmCPU).

Pearson’s correlation coefficients (*r*) were computed to test the relationships for GPC among the growing seasons and with the BLUEs. 

To measure the genetic influences of GPC, the broad-sense heritability (*H*^2^) was estimated according to the following formula:$$H^{2}=\frac{\sigma_{G}^{2}}{\left(\sigma_{G}^{2}+\frac{\sigma_{E}^{2}}{nE}\right)}$$
where $\sigma_{G}^{2}$ is the genotype variance, $\sigma_{E}^{2}$ is the variance of the residual and *nE* is the number of the of environments (growing seasons).

To estimate the heritability of GPC in each environment, the same formula was used, where $\sigma_{G}^{2}$ is the genotype variance, $\sigma_{E}^{2}$ is the variance of the residual, but *nE* is the number of the replications in a given environment.

All phenotypic data analyses were accomplished using STATISTICA 14 [26].

### 2.4. Phylogenetic Analysis

The wheat panel has been previously genotyped using the 90 K iSELECT SNP array by the SGS Institut Fresenius GmbH TraitGenetics Section, Gatersleben, Germany. To perform phylogenetic analysis, a dataset of 17,093 polymorphic SNPs for the 255 wheat accessions was used as input data in the R package “phyclust” [27]. The phylogenetic tree was constructed by the Unweighted Pair Group Method with Arithmetic Mean (UPGMA) algorithm [28].

### 2.5. Genome-Wide Association Study (GWAS) Analysis

The population was previously described and more information about the filters applied to the SNP markers database, population structure, and linkage disequilibrium (LD) is provided in [29]. 

GWAS was performed by applying the FarmCPU algorithm through GAPIT 3 (Genomic Association and Prediction Integrated Tool) in R. This algorithm was implemented to avoid any false-negative associations and to control the false positive ones in accordance with [15]. The association signals by GWAS were calculated for phenotypic data of each growing season and BLUEs. On the resulting Manhattan plots, the threshold of statistically significant QTNs (quantitative trait nucleotides) or MTAs (marker-trait associations) was set above the false discovery rate (FDR) at the significance level of 0.01 (−log_10_
*p*-values > FDR) calculated for each growing season separately as per [15]. Important *p*-value distributions (expected vs. observed −log_10_ (*p*-values)) were shown with a quantile–quantile plot. 

### 2.6. Candidate Gene Search

LD was calculated and defined by Schierenbeck et al. [29], and was used here as interval for detecting the candidate gene(s). High-confidence putative candidate genes were predicted based on significant SNPs, which passed the FDR threshold, and their extension regions within the LD interval (±2 Mbp) using the reference genome sequence of cv. Chinese Spring by blasting against IWGSC RefSeq annotation v1.1. [30]. Three web-based platforms (WheatIs, EnsemblPlants, and Persephone) were used to obtain the gene annotations, gene ontologies (GO) and InterPros, and descriptions for the potential candidate genes (https://urgi.versailles.inrae.fr/wheatis/; http://plants.ensembl.org/Triticum_aestivum/Info/Index; https://web.persephonesoft.com/?data) (accessed on 10 June 2022).

## 3. Results

### 3.1. Phenotypic Assessment

Large phenotypic variation was observed for GPC across the whole dataset from three crop growing seasons within the panel of 255 wheat accessions (Figure 1A,B, Appendix A). The GPC values presented as a percentage (%) in the accessions for the individual growing seasons ranged from 5.7 to 18.1%. The accession values averaged over the three growing seasons ranged from 8.6 to 16.4% with a total average of 11.3% ± 0.09. The widest phenotypic variation of GPC was observed in 2016 growing season followed by 2017 (Figure 1B). To assess the population phenotypic variability across the growing seasons, we calculated the deviations from the corresponding annual average GPC (in %); then, for each accession, the deviations over the three seasons were averaged. Genotypic effects varied along with the environmental gradient; the variance ranged from 0.024 (2018) to 0.031 (2017) (Appendix A). The population variance over the growing seasons ranged from 0 to 0.147 (Appendix A). Some accessions displayed consistently high GPC across the environments, for instance, Bezenchukskaya 380, Rannaya 12, Moskovskaya 40 (Russia) and ZG K 238/82 (Croatia) (high positive deviation from the average GPC, low variance), whereas others such as Wagrein (Austria), was inconsistent across the crop seasons although with a high average GPC (high positive deviation from the average GPC, high variance), or had a consistently low GPC, for instance, Folke (Sweden), Durin and GSA 3 (France) (high negative deviation from the average GPC, low variance) (Appendix A). 

The factorial analysis of variance showed significant effects of Genotype, Growing season and their interaction (Appendix A). To eliminate the environmental impact, BLUEs were calculated for each accession across the growing seasons by assuming the genotype as a fixed effect and the growing season as a random effect. The distribution of the BLUEs across growing seasons also showed wide genotypic variation in the range 10.4–12.6% (Figure 1A,B, Appendix A). The three accessions with the highest average GPC based on BLUEs were all samples of the Russian cultivar Bezostaya 1 coming from different sources (genebank collections) (Figure 1C). Accessions from Northern, Western and most of Central Europe, as well as Canada had average protein values lower than the BLUE averaged over the entire panel, whereas accessions from Eastern Europe, part of Central Europe (Hungary and Austria), Asia, and Australia had average protein values higher than the average BLUE (Appendix A). 

As expected for a quantitative trait, the distribution of GPC means of the 255 accessions followed normal curves across the three growing seasons, with a large number of accessions covering protein contents within the range of 10–12% for each environment (Figure 2A). Low to moderate positive Pearson’s correlation coefficients, denoted as *r*, were calculated for GPC across the growing seasons and with the BLUEs (Figure 2B).

All genotypes were grouped into three clusters of low (mean 10.7%), medium (mean 11.2%) and high (mean 11.8%) GPC based on BLUEs, and comprising 77, 80, and 98 accessions, respectively (Figure 3A, Appendix A). Figure 3B shows the proportion of the entries of low-, medium-, and high-grain protein in the pools from each country. 

### 3.2. Phylogenetic Analysis

The phylogenetic tree analysis of the 255 diverse wheat accessions was based on 17,093 polymorphic SNP markers. The accessions were grouped into six distinctive clusters related mainly to their geographical origin (Appendix A). The largest cluster (I) consisted of 99 accessions predominantly from the former Soviet Union (Russia, Kazakhstan and Ukraine), few accessions from Bulgaria, and a sub-cluster of US cultivars. Clusters III and V comprised almost exclusively accessions from North America and Scandinavia, respectively. The accessions from South-Eastern Europe formed the core of cluster IV, whereas those from Central and Western Europe dominated in cluster VI. Cluster II comprised 16 accessions of mixed origin.

### 3.3. Association Mapping Analysis

For GWAS, we used the genotype data for the 255 accessions that were already available from the 90 K iSELECT SNP technology (SGS Institut Fresenius GmbH TraitGenetics Section, Gatersleben, Germany), and described by Schierenbeck et al. [29]. Although high broad-sense heritability estimates were observed for GPC in the individual growing seasons (*H*^2^ = 0.79 in 2016, 0.68 in 2017, and 0.78 in 2018) (Appendix A), on average *H*^2^ = 0.79, the analysis of variance revealed significant genotype × environment interaction (Appendix A). Therefore, we performed GWAS for GPC first for each growing season, followed by analysis based on BLUEs.

In total, 22 significant QTNs or MTAs for GPC were detected across the three growing seasons and based on BLUEs at −log_10_ (*p*-value) > FDR (Table 1, Figure 4A). The FDR threshold was at −log_10_ > 4.2, and therefore, the number of spurious associations was reduced. The 22 significant QTNs were located on 14 chromosomes: 1B (1 in 2018), 1D (1 with BLUEs), 2B (1 in 2016), 3A (2 with BLUEs, 2 in 2016, 2 in 2017, 1 in 2018), 3B (1 with BLUEs, 1 in 2017, 1 in 2018), 3D (1 with BLUEs), 4A (1 in 2018), 4B (1 with BLUEs), 5A (1 with BLUEs), 5B (1 in 2018), 6A (1 in 2016), 6B (1 in 2018), 7A (1 in 2018) and 7B (1 in 2016) (Table 1, Figure 4A). Most of the QTNs showed negative additive effect reducing the GPC value (ranging from −0.86 to −0.30%. The findings highlighted four SNPs (*Excalibur_rep_c112060_100* on 7B, *Excalibur_c39248_485* and *Excalibur_c13709_2568* on 3A, and *wsnp_Ex_c16432_24932860* on 5B) with positive additive effect ranging from 0.32 to 0.70% which can be used to increase GPC.

Three shared marker associations were found: two on chromosome 3A (markers *Excalibur_c13709_2568*, detected in growing season 2018 and based on BLUEs, and *wsnp_Ku_c7811_13387117*, detected in 2017 and with BLUEs), and one on chromosome 3B (*BS00062734_51*, detected in growing seasons 2017 and 2018, and based on BLUEs) (Table 1, Figure 4A). These QTNs could be considered as stable ones. Although no shared SNP was found with growing season 2016, a significant marker (*Ku_c56370_1155*) was detected on chromosome 3A at a position very close (2.8 Mbp) to the stable marker *Excalibur_c13709_2568*. The common SNPs showed contrasting effects on the GPC. The marker *Excalibur_c13709_2568* had positive influence that can be used to improve the protein content in wheat grains. In contrast, the two other shared SNPs showed a negative effect on GPC (Table 1).

Applying FarmCPU model using 90 K SNPs for the association analyses, we detected strong association signal within several interesting genomic regions. The Q-Q plots supported these findings and therefore, we used significant SNPs/QTNs that were obtained with BLUEs and verified in at least one crop season in further analyses for detecting candidate genes.

### 3.4. Candidate Genes Associated with Significant SNPs

Applying the criteria to select the most significant SNPs/QTNs, 127 high-confidence genes in total were found based on the markers’ physical position and within the LD interval (±2 Mbp) determined by the LD decay. There were 19, 74 and 34 candidate genes, located within the LD screen of the three significant shared QTNs on chromosomes 3A (markers *Excalibur_c13709_2568* and *wsnp_Ku_c7811_13387117*), and 3B (marker *BS00062734_51*), respectively. The found putative candidate genes included 122 genes, 9 probable transposons, 1 pseudogene; 4 were undefined. Of these, 113 were indicated to have molecular functions, 61 were biological processes, and 13 were involved in cellular components, as shown in Figure 5. Detailed descriptions of these candidate genes are summarized in Appendix A.

The top five ontologies attributed to molecular function were: protein binding, nucleic acid binding, hydrolase activity, ATP and GTP binding, and metal ion binding. Similarly, for biological processes, the top five categories were: metabolic process (including carbohydrate, DNA, and amino-acids metabolism), biosynthetic process, oxidation-reduction process, regulation of transcription, and protein post-translational modifications. The key cellular component where these molecular function and biological process ontologies functionally comprised the biological system were membranes and their integral constituents. The complete list of molecular function, biological process, and cellular component ontologies is tabulated in the Appendix A.

Most of the genes related to molecular functions and biological processes which were identified through GO analysis, code for proteins that were categorized into the following classes: (1) transporter proteins; (2) transcription factors; (3) proteins involved in post-translational modifications; (4) proteins involved in biosynthesis of macromolecules; (5) protective proteins; (6) structural proteins; (7) proteins bound up in metal ions transport and homeostasis. Putative key genes pertaining to different processes related to the accumulation of protein during grain development that possibly are differentially expressed in the population are presented in Table 2, and the most prominent ones are discussed in the next section.

## 4. Discussion

The grain protein content is one of the main determinants of the nutritional and baking quality of wheat. More than 80% of their final N content wheat plants accumulate before anthesis as leaf plastidial proteins [31]. During the grain-filling period, which coincides with leaf senescence, the degradation of plastidial proteins, mainly Rubisco, provides the N available for remobilization from leaves to ears to be used for protein synthesis. Due to the reverse relationship between grain yield and quality, the selection of wheat cultivars of high yield influences the protein levels in the grain despite the better economic return in areas where cereals contribute to the largest daily dietary intake. The resulting deficiency in proteins and essential micronutrients in the cereal-based diet known as “hidden hunger” is recognized as one of the severe health issues for pregnant women and young children, particularly in developing countries. Enhancing the nutritive levels of wheat plants using molecular breeding approaches relies on the knowledge of phenotypic variability and the genetic dissection of the nutrition associated traits.

### 4.1. Phenotypic Variation for Grain Protein Content

Growing wheat with high grain protein begins with selecting of appropriate genotypes followed by management practices [32]. Therefore, the information about genetic diversity for this trait is important. Since GPC is highly influenced by the interaction between the environmental and genetic factors, our study was designed to consider the impact of these main factors to estimate GPC using world-wide diverse genotypes across three consecutive growing seasons. In the current study, the phenotypic data analyses revealed considerable variation for GPC. The obtained high heritability and the moderate correlations observed among growing seasons demonstrated consistency of GPC across the environments and suggested a substantial genetic component to the variation of GPC. Comparable heritability estimates were also reported in previous studies [14,33].

Environmental factors, in particular reduced precipitation and rising temperatures at critical vegetation phases are a major pressure on wheat production with impact on both grain yield and GPC [34]. During the three growing seasons, the average monthly temperatures were, in general, close to or higher than the climate norm for the seed-providing experimental field (Appendix A). However, in 2018, from heading to the end of grain filling phase (May–July), the precipitation was significantly lower compared to the climate norm, coupled with higher average monthly temperatures (up to 4 °C deviation), which is suggestive of prolonged drought. During the grain filling phase (July), the precipitation was considerably reduced in 2016, whereas in 2017 extreme rainfall values were recorded (Appendix A). These fluctuations among the three crop seasons might also account for the observed variation in the protein content in the studied set of 255 wheat accessions. Terminal drought and especially the prolonged drought period have been shown to increase GPC due to elevated N uptake and proline synthesis when plants suffer from water insufficiency [34]. In our study, the average protein values for 2016 (11.29%) and 2018 (11.93%), when drought periods were suggested during the late vegetation, were significantly higher compared to 2017 (10.60%) (*p* < 0.001; Appendix A). The Genotype × Growing season interaction was significant (Appendix A), and the genotypic effects varied along with the environmental gradient, in particular, variation of different extent was observed in protein levels in accessions of various geographic origin across the environments (Appendix A). The results on the phenotypic variability for protein content is largely in accordance with the spatial distribution of the wheat accessions demonstrated also by the phylogenetic tree (Appendix A). The 255 diverse accessions were clustered into six distinctive groups mainly based on their geographic origin. For example, the largest cluster (I) comprised accessions from Russia, Ukraine, Kazakhstan, and Bulgaria (Appendix A). The accessions pools from these countries have large fractions of high protein entries (Figure 3). The obtained outcome is in line with the grain quality reports for the wheat germplasm from these countries [18,35,36,37,38,39,40,41]. In addition, we assume that the accessions grouped, to some extent, according to their genealogy. For example, based on known pedigree information, Russian cultivars especially Bezostaya 1, Avrora, Skorospelka 35, Kavkaz, Mironovskaya 808, Rannaya 12, Krasnodarskij karlik, as well as Ukrainian cultivars, such as Odesskaja 86, Obrij, Khersonskaja 552, Yuzhnaya Zarya, etc., have been broadly used as parents for hybridization by the breeders in Bulgaria since the 1950s [42]. The Russian cultivar Bezostaya 1 demonstrated the highest grain protein based on BLUEs, and this result was consistent in the three entries maintained at different seed gene banks. This partly explains the obtained high values of the entries from the aforementioned European germplasm and their positioning in the same cluster.

### 4.2. Quantitative Trait Nucleotides

Applying GWAS approach in the diverse population with dense SNPs using GPC values (%), empowered us to detect 15 significant QTNs distributed on 10 chromosomes, whereas GWAS performed with BLUEs identified 7 significant QTNs on 6 chromosomes. The most significant QTNs were mapped on chromosomes 1B, 1D, 2B, 3A, 3B, 3D, 4A, 4B, 5A, 5B, 6A, 6B, 7A, and 7B, which were mostly detected in single environments (crop growing seasons). These results are in agreement with previous studies reporting marker associations with GPC using different wheat populations. Thus, using a global core collection, made up of 372 accessions from 70 countries, 33 markers spread over 15 wheat chromosomes (1A, 1B, 1D, 2A, 2B, 2D, 3B, 3D, 4B, 5B, 5D, 6A, 6B, 6D, and 7B) were detected in association with GPC [43]. The GWAS approach applied to CIMMYT breeding material identified new GPC marker associations on chromosomes 6A and 7A [44]. More recent association mapping studies revealed novel QTL regions on chromosomes 2B, 5A, 5D, and 7B in a collection of local and modern Iranian accessions [45]. In wild emmer, Liu et al. [16] identified molecular markers grouped into two novel GPC-linked QTL regions on chromosomes 2B and 7B, using a worldwide collection of 486 wheat accessions from China, Europe, and America. A recent study detected genomic regions associated with GPC on chromosomes 2B and 6A, using a diverse panel of 372 European accessions [46]. In total, 10 GPC-linked QTLs represented by 45 SNPs were mapped on chromosomes 1A, 1B, 2B, 3A, 3D and 6A, using a set of 486 accessions from worldwide regions, especially China [47]. Alemu et al. [17] reported many chromosomal regions on 2A, 3B, 3D, 5B, 6A, 6B and 7B as the source of highly significant MTAs for GPC based on GWAS analysis in 802 elite spring breeding lines. The contribution of loci on chromosomes 1A, 2A, 2B, 3A, 3B, and 6B for protein content was confirmed in a GWAS study on UK wheat genotypes [48], whereas genomic regions on chromosome 6A were recently confirmed in a study on Russian spring cultivars [18]. QTNs for GPC under limited water supply were located on chromosomes 1B, 2A, 2B, and 6B [49], of which the QTL 6B.7 was co-mapped with the previously cloned gene *Gpc-B1* associated with high grain protein [20]. In total, 16 significant MTAs for GPC on chromosomes 1A, 1B, 1D, 2B, 3B, 4B, 5A, 5B, 5D, 6A, 6B and 7A have been identified in a set of 280 diverse bread wheat genotypes [50]. QTL hotspots harboring a number of significant MTAs for protein content have been recently reported on chromosomes 3B (13), 5A (2), 5B (6) and 7D (2) in a population of Mediterranean landraces [51].

A recent meta-analysis of GPC-associated QTLs in hexaploid and tetraploid wheat conducted on 48 linkage-based mapping studies identified stable QTLs (meta-QTLs) and hotspots on all wheat chromosomes, except for 1D and 4D, of which 19 meta-QTLs and 2 hotspots on 14 chromosomes were shown to be co-localized with significant SNP-based MTAs retrieved from association mapping studies [52].

The significant QTNs identified in the current study are potentially promising for early selection to improve the marker-assisted selection of wheat grain quality. However, the potential use of marker associations may be questioned considering the variation in their stability and therefore, requires multi-environmentally verified ones [53]. Our results highlighted three stable QTNs: one on 3B that was consistent in three environments (2017, 2018 and BLUEs), one on 3A detected in 2017 and with BLUEs, and another one on 3A verified in two environments (2018 and BLUEs). No shared marker was found with crop season 2016, indicating that due to the environment, there are different genetic SNPs controlling GPC. The consistent SNP on chromosome 3B (*BS00062734_51*) is located at the position of 545,069,014 bp on the IWGSC RefSeq v1.1 map and is, therefore, distant from the two significant SNPs identified on the same chromosome in the study by Alemu et al. [17] within the intervals 822,819,158–822,820,535 bp, and 795,567,173–795,572,772 bp, as well as far from the five significant SNPs on 3B reported by Leonova et al. [18], altogether located between 29,356,100 and 493,122,600 bp. It is also far from the reported GPC-SNPs on chromosome 3B (*wsnp_Ex_c20652_29734133* at 292,024,034 bp, and *RAC875_c58159_989* at 564,248,743 bp [48], and *AX-94746929* at 800,933,346 bp [50]). The positions of the two stable SNPs, identified in the present study on chromosome 3A (*Excalibur_c13709_2568* at 518,897,423 bp and *wsnp_Ku_c7811_13387117* at 714,304,435 bp), are far from the significant GPC-QTL peaked by *wsnp_Ku_c30545_40369365* at 363,458,708 bp [48], as well as far from the two significant SNPs on 3A detected by Leonova et al. [18] in single crop seasons (located at 20,019,900 and 739,397,100 bp). In addition, both are distantly located from the multi-environmentally verified SNPs mapped to the 3A chromosome region between 191,530,793 and 484,643,825 bp [47]. Moreover, the location comparisons of the stable QTNs identified in our study with those of the three meta-QTLs on chromosome 3A and the two GWAS-validated meta-QTLs on chromosome 3B [52] also imply that these are different loci. This suggests that chromosomes 3A and 3B harbor one or more genomic regions associated with GPC that had not been reported yet. Therefore, we can claim with reasonable certainty that the significant QTNs/SNP associations with GPC on chromosomes 3A and 3B identified in the present study are novel.

Interestingly, the most promising and stable QTNs in our study located on 3A and 3B were found to be strongly associated with other human health components such as macro- and microminerals and other grain yield components [9,19,23], indicating that these QTNs can be useful for the simultaneous improvement not only of protein contents but also for minerals and grain yield.

### 4.3. Candidate Genes

We selected the three significant QTNs obtained with BLUEs on chromosomes 3A and 3B, that were shared with at least one individual growing season and, thereby were considered as stable ones, to pull out putative genes that are potentially responsible for wheat GPC. Using the latest IWGSC RefSeq annotation v1.1, we retrieved 127 high-confidence GPC-related candidate genes in the vicinity (±2 Mbp) of the stable QTNs. Below, we discuss some of the most prominent potential candidate genes encoding for different categories of proteins.

#### 4.3.1. Transporters

Protein synthesis during the grain-filling phase needs abundant supply of amino acids which are remobilized from the vegetative tissues. The transport of amino acids across membranes and translocation from source to sink is mediated by amino acid transporter proteins. Here, we found two candidate genes in chromosome 3A (*TraesCS3A02G484600* and *TraesCS3A02G484900*) whose product is an amino acid permease (Table 2 and Appendix A), a transmembrane protein from the Amino Acid/Auxin Permease family (AAAP). Amino acid transporters are shown to be highly expressed in wheat grain cells and regulate the accumulation of amino acids in grain [54]. Recently, Jin et al. [55] reported on the cloning of three homoeologues of *TaAAP6* (on 3A, 3B and 3D) from *OsAAP6* (a putative amino acid permease gene) by homology cloning, and showed that *TaAAP6-3B* was a regulator of GPC, and its favored allele *TaAAP6-3B-I* has clear potential for application in wheat breeding for enhanced GPC. Based on recently published results, transporters including transporter proteins are expected to have a potential role in grain micronutrient accumulation [9,19,23].

#### 4.3.2. Proteins, Implemented in Transcriptional and Post-Translational Regulation

The output of our candidate gene search included three genes in chromosome 3A conferring NAC domain-containing proteins (Table 2 and Appendix A). Recent transcriptomic analyses in wheat grains reported the expression of several transcription factors, including members of the NAM/ATAF/CUC (NAC) family [56,57]. The wild allele of a NAC gene (NAM-B1, NO APICAL MERISTEM) was reported to encode a transcription factor that accelerates senescence and increases nutrient remobilization from leaves to developing grains [20]. Moreover, the authors reported on positional cloning of *Gpc-B1* gene (NAM-B1), which significantly increases the levels of GPC, Fe and Zn. *NAM* genes have been recently reported to regulate the transcriptional changes during early leaf senescence in wheat including genes associated with nitrogen transport [58]. A GPC-associated SNP on chromosome 5D was found to encode a NAC domain superfamily protein (gene *TraesCS5D02G537600*) as well [50]. A recent study reported an endosperm-specific transcription factor TaNAC019 that regulates the accumulation of grain prolamins (glutenin) and starch, providing a candidate gene for breeding of wheat with improved grain quality [59]. Putative candidate genes for NAC transcription factors and transmembrane proteins were found to play a critical role in the Fe accumulation in wheat grains [8].

In our study, we found three post-translational putative genes in chromosome 3A with GO-identified biological process “protein phosphorylation”, molecular function “protein kinase activity” and functional annotation serine/threonine protein kinase or receptor-like kinase (Table 2 and Appendix A). Genes annotated as protein kinases have shown their potential role in macro- and micronutrients accumulation in wheat grain, as well [23]. One putative gene, *TraesCS3A02G488100*, detected on chromosome 3A, was associated with the biological process “protein glycosylation” and molecular function “acetylglucosaminyltransferase activity” encoding for a glycosyl transferase. Candidate genes coding for proteins with acetylglucosaminyltransferase or kinase activity have been predicted in *Aegilops tauschii* (the D-genome progenitor of bread wheat) genomic regions harboring grain micronutrient (Fe and Mn) MTAs [33].

Our candidate gene search identified two putative genes involved in another post-translational modification—protein ubiquitination: one gene for E3 ubiquitin-protein ligase and another one for the E2 ubiquitin-conjugating enzyme, both on chromosome 3A (Table 2 and Appendix A). We also identified six putative genes encoding F-box proteins on chromosomes 3A (5) and 3B (1) (Table 2 and Appendix A). This is in accordance with the reported high-confidence candidate gene (*TraesCS1B02G046800*) associated with GPC in wheat governing a F-box-like domain superfamily protein on chromosome 1B [52]. The F-box domain-containing proteins are key players in the ubiquitin-mediated protein turnover. Transcriptome analysis during various wheat developmental stages revealed that some F-box genes were specifically expressed at the seed developmental stages [60]. Interestingly, the putative candidate gene *TraesCS3A02G389000* linked to the strongest SNP marker (*AX-95002032*) on chromosome 3A associated with grain Fe concentration in wheat also was found to encode a F-box protein [19]. This observation hints at a possible link between protein and mineral homeostasis.

The stable significant SNP (*BS00062734_51*) on chromosome 3B identified in this study is located within a gene (*TraesCS3B02G339000*) that encodes for a plastidial lipoyl synthase (Table 2 and Appendix A), a protein responsible for the biosynthesis of lipoic acid and further involved in a rare but highly conserved lysine post-translational modification, known as protein lipoylation [61]. Interestingly, *BS00062734_51* was previously reported as the most significant marker in a QTL region on 3B for grain yield and duration of pre-anthesis sub-phases in a set of 210 winter genotypes selected from the GABI_WHEAT population [62].

#### 4.3.3. Proteins, Involved in the Biosynthesis of Other Storage Materials

The biosynthesis of carbohydrates (mostly starch) is one of the most important metabolic processes in developing wheat grains besides the synthesis and accumulation of storage proteins. Wheat reserve starch is synthesized by the enzymatic machinery in the endosperm cells. Key proteins in the starch biosynthesis pathway have been recently identified [63]. In the current work, we identified a candidate gene on chromosome 3A (*TraesCS3A02G289800*), encoding PROTEIN TARGETING TO STARCH (PTST) (Table 2 and Appendix A)—a product that is among the 26 essential proteins in a starch biosynthesis model for wheat endosperm [63]. The suggested functional role of the wheat protein TaPTST1, an *Arabidopsis thaliana* AtPTST1 homolog, in starch metabolism targets the Granule-Bound Starch Synthase I (GBSSI) to starch granules in wheat endosperm [64]. Major starch-biosynthesis-related genes were shown to maintain high expression levels during the grain-filling stage in compliance with the high rates of starch accumulation [57,63,64]. Interestingly, the here-suggested candidate gene *TraesCS3A02G289800* for PTST is located in the vicinity of the significant QTN (*Excalibur_c13709_2568*) with a positive additive effect on GPC (0.54%, Table 1). As a major storage component of wheat endosperm, grain starch content (GSC) correlates positively with grain size and grain yield but shows a negative correlation with GPC. Interestingly, a recent study reported a genomic region on chromosome 6A that controlled both GPC and GSC with opposite allelic effects [46]. Our finding of a putative gene with a positive effect on GPC in a significant SNP-marked genomic region on chromosome 3A, that is involved in starch biosynthesis is suggestive of probable genetic factors that escape the negative GPC-GSC correlation and, therefore, merits further attention.

In the present work, we detected a candidate gene (*TraesCS3A02G289900*) on chromosome 3A hit directly by the significant and consistent SNP (*Excalibur_c13709_2568*), and another one on chromosome 3B (*TraesCS3B02G337100*) (Table 2 and Appendix A) that annotates a metabolic protein–cellulose synthase. This enzyme is active in the plasma membrane-cell wall interface and is involved in the cell wall organization by controlling cellulose biosynthesis. It is also supposedly involved in the defense against pathogens [65]. A recent genome-wide analysis of the cellulose synthase-like (*Csl*) gene family in bread wheat revealed several *Csl* genes, especially concentrated on chromosomes of homoeologous groups 2 and 3 [66].

We found another candidate gene on chromosome 3A (*TraesCS3A02G484800*) hit directly by a significant and stable marker (*wsnp_Ku_c7811_13387117*; Table 1), which encodes for acetyl-CoA-carboxylase (Table 2 and Appendix A). Acetyl-CoA carboxylase (ACCase; EC 6.4.1.2) catalyzes the first committed step in de novo fatty acid biosynthesis. Earlier, the enzyme was shown to be present in both wheat germ and total wheat leaf protein [67].

#### 4.3.4. Protective Proteins

During seed maturation, a series of mechanisms are induced in response to pathogens and environmental stress, in particular desiccation and elevated temperatures. Recent transcriptome and proteome research on wheat developing grain identified several important protein families with protective and detoxification activities [68]. In our study, numerous putative genes were found encoding beta-1,3-glucan hydrolases (beta-glucanases) on chromosome 3A, three genes for glutathione-S-transferases, also on 3A, as well as five genes encoding peroxidases, and two genes for lectin receptor kinases (LRKs) on chromosome 3B (Table 2 and Appendix A). Beta-glucanase expression in seeds plays a key role in the defense against seed pathogens and in the regulation of seed germination and dormancy [69]. Plant glutathione-S-transferases are another class of essential enzymes with detoxification functions during seed development [70], whereas peroxidases play many roles in plant seeds, including the elimination of toxic H_2_O_2_. The LRKs belong to a larger class of receptor-like kinases that play a vital role in the stimulation of immune responses in plants. In wheat, some genes for LRKs (*L-LRK7* and *L-LRK8*) were found to have relatively higher expression in seeds, being extremely upregulated under heat and drought stress [71].

A putative candidate gene *TraesCS3B02G339100* on chromosome 3B that is possibly related to GPC encodes for a subtilisin-like protease (Table 2 and Appendix A). Plant subtilisin-like proteins are involved in both disease resistance and immune priming [72]. Earlier, members of protease families, including subtilisin-like serine proteases were shown to participate in the N remobilization during grain filling [31], and in the breakdown and mobilization of reserve proteins from seeds to cotyledons during germination in wheat [73]. The recently reported functional annotation of the potential candidate genes for GPC in the 6A-chromosome physical region also indicated the presence of subtilisin-like proteases [46].

#### 4.3.5. Structural Proteins

The presence of histones and ribosomal proteins was expected, as they are involved in the nucleosome assembly and ribosomes biogenesis, respectively. Our candidate gene search delivered one putative gene encoding histones on chromosome 3A, and one ribosomal protein encoding gene on chromosome 3B (Table 2 and Appendix A). Recently, Daba et al. [74] reported decreasing abundance of proteins involved in chromatin and ribosomal dynamics during grain development. On the contrary, Bonnot et al. [75] observed high demand for ribosome biogenesis proteins during the grain-filling and early maturation phases. We also identified one putative expansin-coding gene on chromosome 3A (Table 2 and Appendix A). Expansins are important proteins for cell wall loosening, cell enlargement, and extension of grain tissue, and play a vital role in determining grain size [76,77,78]. Recently, three expansin-related genes that were mainly expressed during wheat kernel development were detected on chromosomes 1D, 4B, and 4D [74].

#### 4.3.6. Metal Accumulation and Homeostasis

Based on our investigation, we found six potential genes in chromosomes 3A and 3B that are putatively related to Ca transport, i.e., genes encoding for Ca-channels and Ca-binding proteins (Table 2 and Appendix A). Putative candidate genes for Ca accumulation in seeds, including Ca-channels in addition to other genes that are possibly related to Ca transport, have been recently found by GWAS in a European wheat diversity panel of 353 accessions [7]. A very recent study reported on a histidine-rich Ca-binding protein on chromosome 3B (*TraesCS3B02G019900*) that may be closely related to Ca accumulation in wheat grain [79]. In addition, we detected one potential gene encoding for a metallothionein on chromosome 3A (Table 2 and Appendix A). Metallothioneins are cysteine-rich metal-binding proteins known to coordinate various transition metal ions including Zn, Cd and Cu thus maintaining metal homeostasis and detoxification [80]. These findings suggest that protein accumulation and metal ion transport and homeostasis in wheat grains might share common genetic mechanisms.

## 5. Conclusions

This study showed the importance of improving the grain protein content (GPC) in wheat by GWAS. A wide variation for GPC was found within the studied worldwide panel of 255 bread wheat accessions, along with significant genotypic and environmental effects with high broad-sense heritability estimates. The GWAS analysis conducted on the average BLUE values for GPC from three consecutive crop seasons identified seven significant marker-trait associations (MTAs) on chromosomes 1D, 3A (two MTAs), 3B, 3D, 4B and 5A. The loci on 3A and 3B are stable and novel ones underlying GPC in wheat. The most relevant putative candidate genes in the genomic regions on 3A and 3B included sequences encoding: (1) a subtilisin-like serine protease, with possible participation in N remobilization during grain filling; (2) amino acid permeases, having transporter functions; (3) transcription factors, protein kinases, and ubiquitin-related proteins possibly involved in transcriptional and post-translational regulatory functions; (4) enzymes from the biosynthetic pathways of starch, cellulose, and fatty acids; (5) protective proteins, including beta-gluconases, glutathione-S-transferases, peroxidases, and lectin receptor kinases; (6) histones, ribosomal proteins, and expansins, involved in chromatin, ribosomes, and cell wall dynamics during grain development; (7) proteins related to metal ions transport and homeostasis. Given the complexity of the GPC, many potential genes can be associated with the trait; therefore, this study provided a reference for genetic improvement of efficient wheat grain proteins for human nutrition and quality purposes. Moreover, the identification of nutritional genomic regions with pleiotropic effects on GPC and macro- and microminerals provides guidance for pyramid breeding of cultivars with higher nutritious value.

## Figures and Tables

**Figure 1 foods-12-01399-f001:**
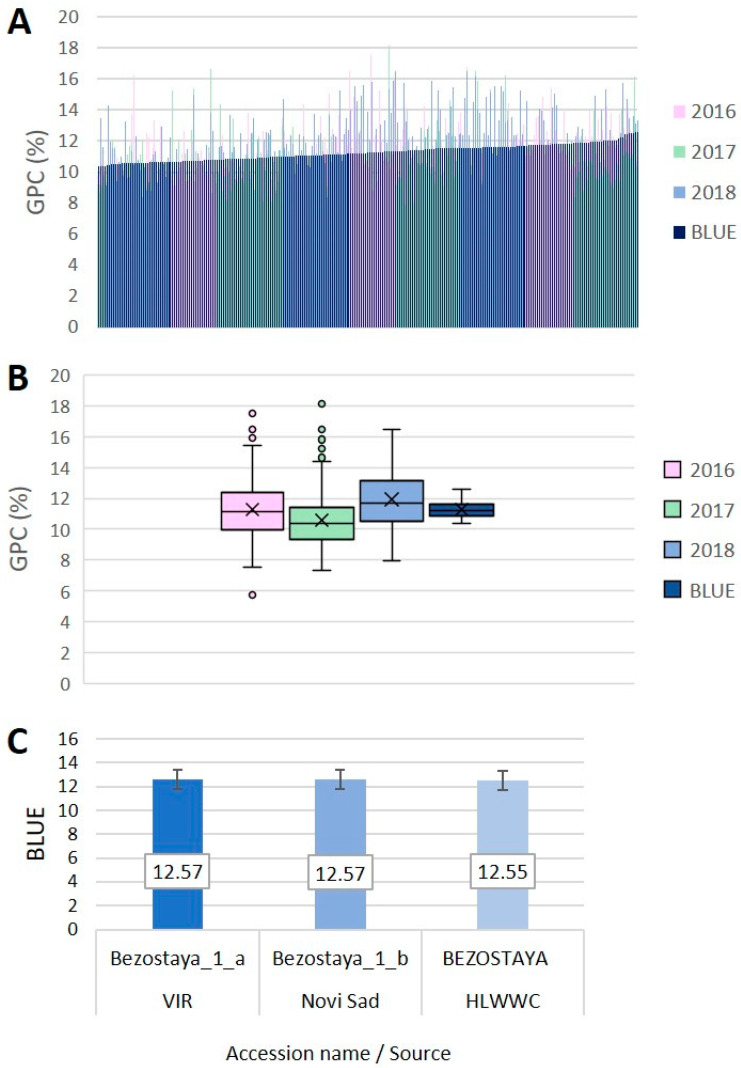
(**A**) Distribution of 255 bread wheat accessions from 27 countries for grain protein content (GPC, %); (**B**) box-plots of GPC (%) in three growing seasons (2016, 2017, 2018), and best linear unbiased estimates (BLUEs); (**C**) top three best accessions based on BLUEs.

**Figure 2 foods-12-01399-f002:**
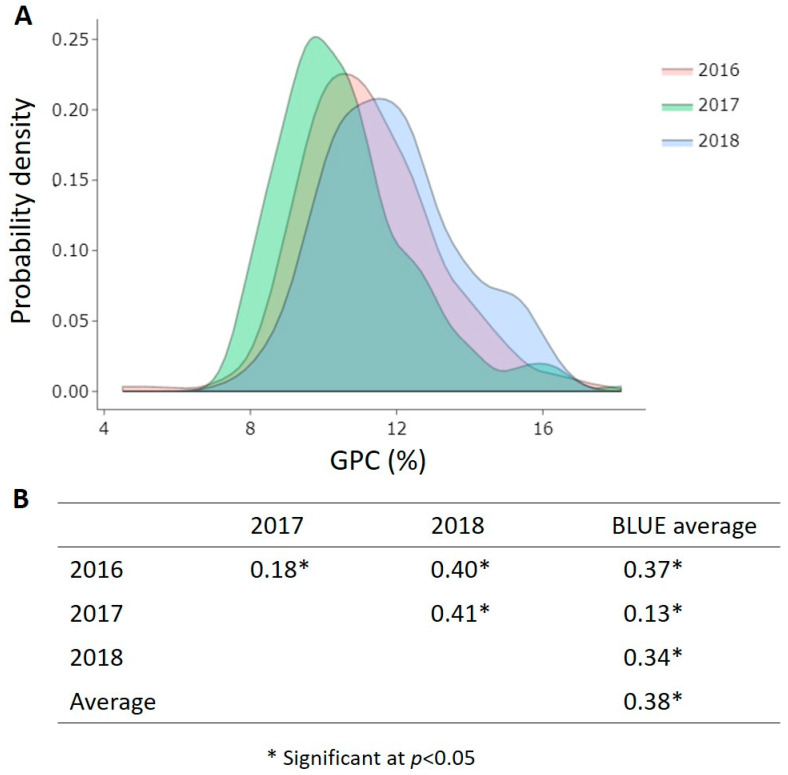
(**A**) Probability density of 255 bread wheat accessions from 27 countries for GPC (%) across three growing seasons (2016, 2017, 2018); (**B**) Pearson correlation coefficient values *r* of GPC among all measured environments and the BLUEs.

**Figure 3 foods-12-01399-f003:**
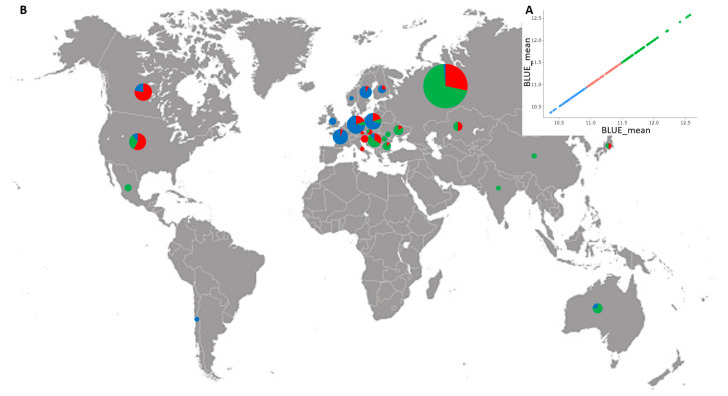
(**A**) Cluster distribution of 255 bread wheat accessions from 27 countries for GPC based on BLUEs averaged over three growing seasons; (**B**) geographic distribution of the wheat accessions. Circles indicate the proportion of accessions with low (blue), medium (red), and high (green) GPC in each country pool.

**Figure 4 foods-12-01399-f004:**
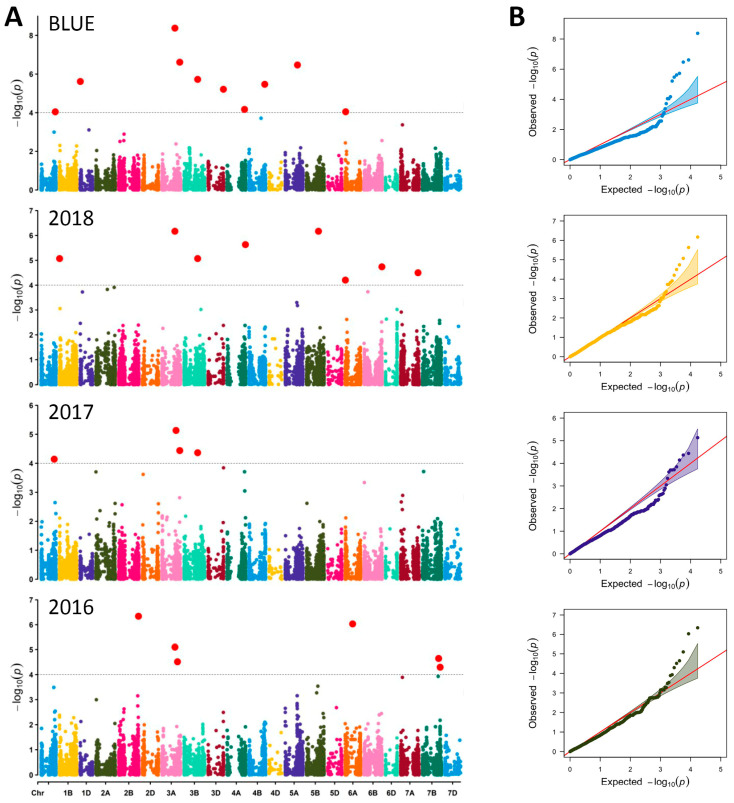
(**A**) Manhattan plots of genome-wide association study for GPC in 255 bread wheat accessions in three growing seasons (2016, 2017, 2018) and BLUEs based on FarmCPU model. The grey color line in the figure corresponds to the threshold of −log_10_ (*p* < 0.001) = 4.2. The significant quantitative trait nucleotides (QTNs) are above the grey color line; (**B**) quantile–quantile plots of GPC; red line depicts the expected values.

**Figure 5 foods-12-01399-f005:**
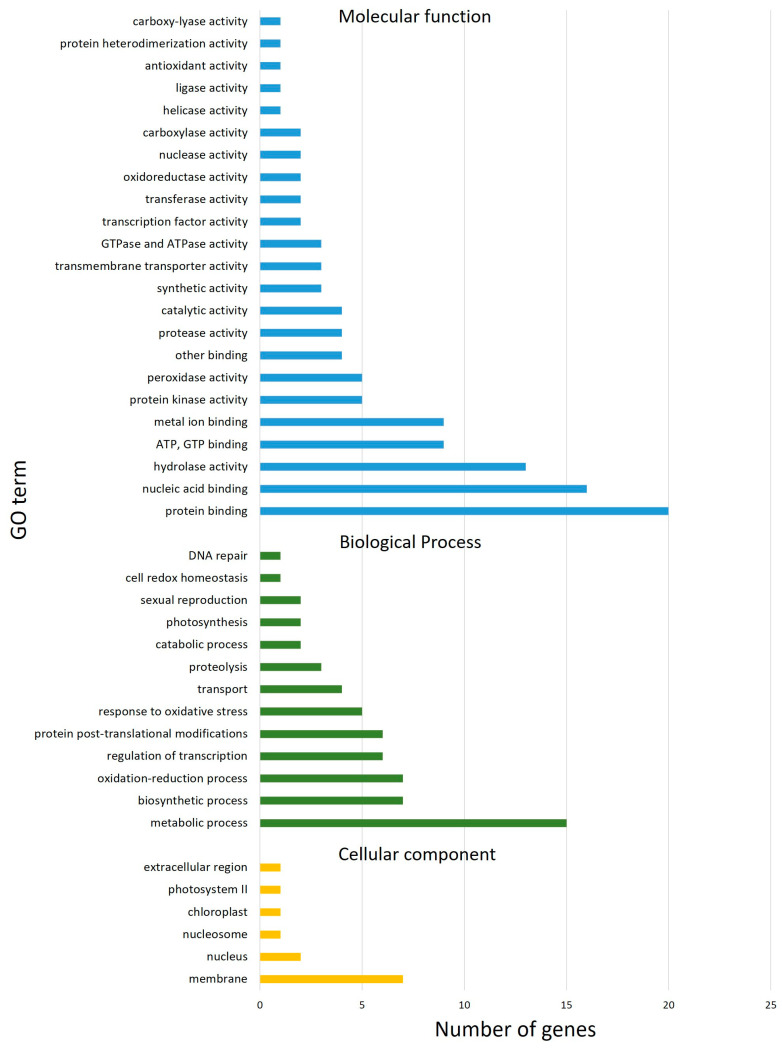
Gene ontology (GO) enrichment analysis.

**Table 1 foods-12-01399-t001:** Significant single nucleotide polymorphism (SNP) markers for grain protein content (GPC) in a diverse bread wheat population of 255 accessions, identified by the fixed and random model circulating probability unification (FarmCPU) model at the threshold of −log_10_ (*p*-value) > FDR (False Discovery Rate).

	SNP	Chr.	Position (bp)	Allele	Effect	*p*-Value	−log_10_
	GPC_BLUE						
1.	*Ex_c67198_1838*	1D	10,259,568	T/C	−0.385547705	2.42 × 10^−6^	5.616184634
2.	*Excalibur_c13709_2568*	3A	518,897,423	A/G	0.543011247	4.20 × 10^−9^	8.37675071
3.	*wsnp_Ku_c7811_13387117*	3A	714,304,435	A/G	−0.391805162	2.43 × 10^−7^	6.614393726
4.	*BS00062734_51*	3B	545,069,014	A/G	−0.323156108	1.89 × 10^−6^	5.723538196
5.	*Kukri_c11944_436*	3D	609,621,771	T/C	−0.318723447	6.13 × 10^−6^	5.212539525
6.	*wsnp_Ra_c1146_2307483*	4B	630,470,822	T/G	−0.304930639	3.38 × 10^−6^	5.4710833
7.	*BobWhite_c6759_365*	5A	488,262,559	T/C	−0.354037284	3.39 × 10^−7^	6.469800302
	GPC_2018						
8.	*Excalibur_c13709_2568*	3A	518,897,423	A/G	0.570293042	8.48 × 10^−6^	5.071604148
9.	*BS00062734_51*	3B	545,069,014	A/G	−0.323156108	2.33 × 10^−6^	5.632644079
10.	*Kukri_c37738_417*	1B	5,519,895	T/C	−0.570	8.48 × 10^−6^	5.632644
11.	*Tdurum_contig46583_2146*	4A	738,781,353	T/C	−0.418	2.33 × 10^−6^	5.071604
12.	*wsnp_Ex_c16432_24932860*	5B	483,000,817	T/C	0.696768643	6.76 × 10^−7^	6.170053304
13.	*Excalibur_c28759_914*	6B	716,259,097	A/G	−0.445651367	1.83 × 10^−5^	4.73754891
14.	*wsnp_Ex_c22547_31738007*	7A	694,546,597	T/C	−0.324192625	3.17 × 10^−5^	4.498940738
	GPC_2017						
15.	*wsnp_BE490651A_Ta_2_2*	3A	565,466,354	T/C	−0.861806961	7.35 × 10^−6^	5.133712661
16.	*wsnp_Ku_c7811_13387117*	3A	714,304,435	A/G	−0.478132901	3.64 × 10^−5^	4.438898616
17.	*BS00062734_51*	3B	545,069,014	A/G	−0.421626697	4.32 × 10^−5^	4.364516253
	GPC_2016						
18.	*Excalibur_c12675_1789*	2B	788,655,682	A/G	−0.402444191	4.57 × 10^−7^	6.3400838
19.	*Ku_c56370_1155*	3A	521,719,100	A/G	−0.355731398	7.96 × 10^−6^	5.099086932
20.	*Excalibur_c39248_485*	3A	624,946,759	A/G	0.377677627	3.07 × 10^−5^	4.512861625
21.	*Kukri_c8274_502*	6A	297,714,501	A/G	−0.726532079	9.30 × 10^−7^	6.031517051
22.	*Excalibur_rep_c112060_100*	7B	646,065,982	T/C	0.316001498	2.26 × 10^−5^	4.645891561

**Table 2 foods-12-01399-t002:** Potential candidate genes underlying GPC based on IWGSC RefSeq annotation v1.1.

**Gene ID**	**Chr:/Position**	**GO Ontology**	**Description**
**Transporter proteins**			
*TraesCS3A02G484600*	chr3A:714289919..714291827 (−strand)	BP: amino acid transmembrane transport; MF: amino acid transmembrane transporter activity; CC: membrane	Amino acid permease
*TraesCS3A02G484900*	chr3A:714415263..714423339 (−strand)		
**Transcription factors**			
*TraesCS3A02G485400*	chr3A:7145774421..714575250 (+strand)	MF: DNA binding; BP: regulation of transcription, DNA-templated	NAC domain-containing protein
*TraesCS3A02G485500*	chr3A:714577793..714578643 (+strand)		
*TraesCS3A02G486500*	chr3A:714650271..714650843 (−strand)		
**Proteins, involved in post-translational modifications**			
*TraesCS3A02G290300*	chr3A:519244293..519255598 (−strand)	MF: protein kinase activity; MF: ATP binding; BP: protein phosphorylation	Receptor-like kinase Serine/threonine-protein kinase
*TraesCS3A02G485600*	chr3A:714588344..714590684 (−strand)		
*TraesCS3A02G486400*	chr3A:714643608..714646388 (+strand)		
*TraesCS3A02G488100*	chr3A:716067049..716070839 (+strand)	BP: protein glycosylation; MF: acetylglucosaminyltransferase activity	Alpha-1,3-mannosyl-glycoprotein 2-beta-N-acetylglucosaminyltransferase
*TraesCS3A02G481900*	chr3A:712621085..712621999 (+strand)	MF: protein binding	E3 ubiquitin-protein ligase E2 ubiquitin-conjugating enzyme
*TraesCS3A02G487000*	chr3A:714787620..714789640 (+strand)		
*TraesCS3A02G484000*	chr3A:713962301..713963870 (−strand)	MF: protein binding	F-box family protein
*TraesCS3A02G483900*	chr3A:713945850..713955055 (−strand)		
*TraesCS3A02G482600*	chr3A:713210615..713212190 (+strand)		
*TraesCS3A02G485800*	chr3A:714595534..714599278 (+strand)		
*TraesCS3A02G487100*	chr3A:714939243..714940190 (+strand)		
*TraesCS3B02G340400*	chr3B:547054749..547060421 (−strand)		
* *TraesCS3B02G339000*	chr3B:545066258..545069534 (+strand)	MF: catalytic activity; BP: lipoate biosynthetic process; CC: chloroplast; MF: lipoate synthase activity; MF: iron-sulfur cluster binding; MF: 4 iron, 4 sulfur cluster binding	Lipoyl synthase
**Proteins involved in biosynthesis** **of** **macromolecules**			
*TraesCS3A02G289800*	chr3A:518720116..518723505 (+strand)		PROTEIN TARGETING TO STARCH (PTST)
* *TraesCS3A02G484800*	chr3A:714298564-714310707 (−strand)	MF: catalytic activity; MF: acetyl-CoA carboxylase activity; MF: biotin carboxylase activity; MF: ATP binding; BP: fatty acid biosynthetic process; MF: ligase activity; MF: metal ion binding	Acetyl-CoA carboxylase
* *TraesCS3A02G289900*	chr3A:518895097..518901097 (+strand)	CC: membrane; MF: cellulose synthase activity; BP: cellulose biosynthetic process	Cellulose synthase
*TraesCS3B02G337100*	chr3B:543466729..543471191 (−strand)		
**Protective proteins**			
*TraesCS3A02G483800*	chr3A:713930115..713932275 (−strand)	MF: hydrolase activity, hydrolyzing O-glycosyl compounds; BP: carbohydrate metabolic process	Beta-1,3-glucanase
*TraesCS3A02G483700*	chr3A:713867934..713870026 (−strand)		
*TraesCS3A02G483600*	chr3A:713849212..713849871 (−strand)		
*TraesCS3A02G483100*	chr3A:713622221..713623907 (−strand)		
*TraesCS3A02G483000*	chr3A:713534311..713536049 (−strand)		
*TraesCS3A02G482200*	chr3A:712728151..712729578 (−strand)		
*TraesCS3A02G482000*	chr3A:712698902..712700166 (+strand)		
*TraesCS3A02G481600*	chr3A:712552546..712554003 (+strand)		
*TraesCS3A02G481500*	chr3A:712545116..712546522 (+strand)		
*TraesCS3A02G485200*	chr3A:714535163..714536122 (+strand)		
*TraesCS3A02G486000*	chr3A:714614717..714616126 (+strand)		
*TraesCS3A02G487700*	chr3A:715593874..715594918 (−strand)	MF: protein binding	Glutathione S-transferase
*TraesCS3A02G488200*	chr3A:716097184..716098213 (−strand)		
*TraesCS3A02G488300*	chr3A:716105661..716106534 (−strand)		
*TraesCS3B02G338900*	chr3B:545061480..545063744 (+strand)	MF: protein kinase activity; MF: ATP binding; BP: protein phosphorylation; MF: carbohydrate binding	Lectin receptor kinase, Protein kinase domain; Serine/threonine-protein kinase; Concanavalin A-like lectin/glucanase domain
*TraesCS3B02G338800*	chr3B:545048539..545050613 (+strand)		
*TraesCS3B02G339400*	chr3B:545385674..545385674 (+strand)	MF: peroxidase activity; BP: response to oxidative stress; MF: heme binding; BP: oxidation-reduction process	Peroxidase
*TraesCS3B02G339500*	chr3B:545432618..545433981 (+strand)		
*TraesCS3B02G339600*	chr3B:545493643..545495202 (+strand)		
*TraesCS3B02G339700*	chr3B:545562324..545563678 (+strand)		
*TraesCS3B02G339800*	chr3B:545667818..545669160 (+strand)		
*TraesCS3B02G339100*	chr3B:545179979..545187120 (+strand)	MF: serine-type endopeptidase activity; BP: proteolysis	Subtilisin-like protease
**Structural proteins**			
*TraesCS3A02G484500*	chr3A:714223433..714224075 (+strand)	CC: nucleosome; MF: DNA binding; MF: protein heterodimerization activity	Histone H3
*TraesCS3B02G339900*	chr3B:5457066521..545709391 (+strand)	CC: nucleus; BP: regulation of transcription, DNA-templated	40S ribosomal protein S25
*TraesCS3A02G488000*	chr3A:715973684..715975189 (+strand)	CC: extracellular region; BP: sexual reproduction	Expansin protein
**Proteins involved in metal ions transport and homeostasis**			
*TraesCS3A02G484300*	chr3A:714150436..714156128 (+strand)		Voltage-dependent L-type calcium channel subunit
*TraesCS3A02G488400*	chr3A:716258725..716259719 (+strand)	MF: calcium ion binding	Calcium binding protein
*TraesCS3B02G338600*	chr3B:545008333..545008942 (−strand)		
*TraesCS3B02G338400*	chr3B:544921860..544922297 (−strand)		
*TraesCS3B02G338200*	chr3B:544818884..544819617 (−strand)		
*TraesCS3A02G484700*	chr3A:714292993..714297477 (−strand)	BP: amino acid transmembrane transport; MF: amino acid transmembrane transporter activity; CC: membrane	Metallothionein

* Genes hit by a significant stable QTN.

## Data Availability

Seeds from the bread wheat (*Triticum aestivum* L.) cultivars and lines were kindly provided by the Seed Genebank, Leibniz Institute for Plant Genetics and Crop Plants Research (IPK, Gatersleben), Germany.

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
