# Peer review of "Nutritional Genomic Approach for Improving Grain Protein Content in Wheat"

_foods, 2023, doi:10.3390/foods12071399_

Round 1

Reviewer 1 Report

Dear authors,

Minor corrections and comments can be found from the attached PDF.

Best,

Author Response

Dear Reviewer,

Thank you very much for your constructive comments and suggestions. Please, find below our responses.

  1. Line 27: scan/study

Answer: corrected

  1. Line 118: Would suggest to add phylogeny and cluster analyses, and also analyse the evolution as you have a diverse group of germplasm

Answer: We thank the Reviewer for this suggestion. We performed phylogenetic analysis and included a phylogenetic tree as a supplementary Figure S2. Accordingly, we added brief descriptions of the methodology and the results, as well as a short discussion. Please, see the highlighted text on lines 162-169, 249-258, 404-416.

  1. Line 168 and further places: style of citing some references

Answer: The Reviewer’s remark regarding citing style is taken into consideration. Accordingly, at many places corrections were made (not marked). A check with the latest Journal issues suggested that mentioning of the authors name is also possible. Nevertheless, we tried to keep to minimum the authors names throughout the text.

  1. Lines 269 and 294: quality of Figures 4 and 5

Answer: We agree with the Reviewer’s remark that the quality of figures embedded in the maintext (as per the Journal’s requirements) is poor. We thank the Reviewer for these notes. Accordingly, we did improvements on the figures. They will be submitted as separate files in the required format together with the revised version.

  1. Line 353: Make sure the latest studies on GPC QTN are cited.

Answer: We included a few new references: [48] White et al. 2021; [49] Govta et al. 2022; [50] Krishnappa et al. 2023; [51] Yannam et al. 2023.

  1. Line 414: Need to support your results under this Heading with some latest findings of 2022 or 2023.

Answer: We included a few new references: [64] Sharma et al. 2022; [78] Tillet et al. 2022; [58] Andleeb and Borrill 2023; [50] Krishnappa et al. 2023; [73] Diaz-Mendoza et al. 2019 (earlier work, but relevant)

Sincerely,

Svetlana Misheva

Reviewer 2 Report

The manuscript “Nutritional genomic approach for improving grain protein content in wheat” described the use of GWAS method to identify the trait linked markers related to the grain protein content (GPC) in 255 wheat accessions from 27 countries. Some marker traits associations (MTAs) and candidate genes for controlling GPC were revealed in this study.

Minor errors/Corrections

  1. Page 1, Line 27, GWAS (should be Genome-Wide Association Studies instead of scan)

  2. Page 9, for Figure 5, GO enrichment Analysis, higher resolution of GO terms (Molecular function, Biological process, Cellular component) are necessary because the text is not readable.

Major Concerns 

  1. Genetics and environment interaction plays an important role in regulating wheat grain protein content. Did the authors detect any wheat grain protein content plasticity / variation in fluctuated field conditions (Gatersleben, Germany in this study) over the different geographical origins of seeds although with high broad sense heritability estimates in this case? Did the authors consider the dynamic effects of QTLs that are correlated with grain protein content due to the environmental variances (different field growing conditions for three growing seasons)? Did the authors observe the genetic effects changes along with the environmental gradient?

  1. Are there any abnormal growing conditions such as severe winter weather that occur during the three growing  seasons?

  1. What is the genetic diversity of the wheat accession resources obtained from 5 different continents used in this study, did the authors perform any phylogenetics analysis?

  2. Did the author detect any previously known / identified marker associated with the protein grain content in those wheat accessions used in this study? Are there any detected GPC markers that are consistently found in three growing seasons?

Author Response

Dear Reviewer,

Thank you very much for your constructive questions and suggestions. Please, see the attachment.

Sincerely,

Svetlana Misheva
